# A broadly neutralizing macaque monoclonal antibody against the HIV-1 V3-Glycan patch

Zijun Wang[1†], Christopher O Barnes[2†], Rajeev Gautam[3†], Julio C Cetrulo Lorenzi[1], Christian T Mayer[1], Thiago Y Oliveira[1], Victor Ramos[1], Melissa Cipolla[1], Kristie M Gordon[1], Harry B Gristick[2], Anthony P West[2], Yoshiaki Nishimura[3], Henna Raina[3], Michael S Seaman[4], Anna Gazumyan[1], Malcolm Martin[3], Pamela J Bjorkman[2], Michel C Nussenzweig[1,5], Amelia Escolano[1]*

[1]Laboratory of Molecular Immunology, The Rockefeller University, New York, United States; [2]Division of Biology and Biological Engineering, California Institute of Technology, Pasadena, United States; [3]Laboratory of Molecular Microbiology, National Institute of Allergy and Infectious Diseases, National Institutes of Health, Bethesda, United States; [4]Center for Virology and Vaccine Research, Beth Israel Deaconess Medical Center, Boston, United States; [5]Howard Hughes Medical Institute. The Rockefeller University, New York, United States

**Abstract** A small fraction of HIV-1- infected humans develop broadly neutralizing antibodies (bNAbs) against HIV-1 that protect macaques from simian immunodeficiency HIV chimeric virus (SHIV). Similarly, a small number of macaques infected with SHIVs develop broadly neutralizing serologic activity, but less is known about the nature of simian antibodies. Here, we report on a monoclonal antibody, Ab1485, isolated from a macaque infected with SHIVAD8 that developed broadly neutralizing serologic activity targeting the V3-glycan region of HIV-1 Env. Ab1485 neutralizes 38.1% of HIV-1 isolates in a 42-pseudovirus panel with a geometric mean IC50 of 0.055 μg/mLl and SHIVAD8 with an IC50 of 0.028 μg/mLl. Ab1485 binds the V3-glycan epitope in a glycan-dependent manner. A 3.5 Å cryo-electron microscopy structure of Ab1485 in complex with a native-like SOSIP Env trimer showed conserved contacts with the N332gp120 glycan and gp120 GDIR peptide motif, but in a distinct Env-binding orientation relative to human V3/N332gp120 glycan-targeting bNAbs. Intravenous infusion of Ab1485 protected macaques from a high dose challenge with SHIVAD8. We conclude that macaques can develop bNAbs against the V3-glycan patch that resemble human V3-glycan bNAbs.

*For correspondence:
aescolano@rockefeller.edu

†These authors contributed equally to this work

## Introduction

Over the last decade, characterization of monoclonal antibodies isolated from HIV-1-infected individuals with broad and potent serologic activity against the virus, revealed that bNAbs target several different conserved epitopes on the HIV-1 Envelope spike protein (Env) using multiple mechanisms of binding (*Burton and Hangartner, 2016*). The target epitopes of many bNAbs are unusual because they combine host-derived glycans with protein components of Env. Longitudinal cohort and structural studies demonstrated that bNAb maturation mediated by somatic hypermutation (*Wei et al., 2003*; *Liao et al., 2013*; *Bonsignori et al., 2017*; *Bonsignori et al., 2016*; *Doria-Rose et al., 2014*; *Moore et al., 2012*; *Bhiman et al., 2015*) occurs in part to accommodate the host-derived glycans that shield Env (*Wei et al., 2003*; *Liao et al., 2013*; *Doria-Rose et al., 2014*; *Moore et al., 2012*; *Bhiman et al., 2015*; *Kong et al., 2013*; *Garces et al., 2014*).

One of these epitopes, is the V3-glycan supersite, an oligomannose patch around the V3 base of Env and the potential N-linked glycosylation site at position 332 of the Env gp120 subunit (*Kong et al., 2013*). bNAbs to this site are among the most potent anti HIV-1 antibodies isolated to date and have considerable breadth, moreover, they are among the most frequently elicited upon natural infection with HIV-1 in humans (*Walker et al., 2010*).

A number of structural studies have shown that the family of V3-glycan bNAbs is highly diverse. They target the GDIR motif of Env using multiple angles of approach and therefore accommodate different surrounding glycans (*Kong et al., 2013*; *Mouquet et al., 2012*; *Walker et al., 2011a*). V3-glycan bNAbs use a diverse group of heavy and light chain immunoglobulin genes and only some require the trimeric form of Env for binding (*Longo et al., 2016*).

BNAbs to this site such as BG18 lack some of the rare features observed in other potent bNAbs, for instance the presence of insertions or deletions that are difficult to induce by vaccination (*Freund et al., 2017*).

Therefore, the V3-glycan supersite is a good candidate epitope for anti HIV-1 vaccine design purposes.

The observation that bNAbs arise during natural infection in humans (*Binley et al., 2008*; *Doria-Rose et al., 2009*; *Li et al., 2009*; *Sather et al., 2009*; *Simek et al., 2009*; *Gray et al., 2011*; *Hraber et al., 2014*; *Rusert et al., 2016*), and that they can block SHIV infection in macaques (*Eichberg et al., 1992*; *Emini et al., 1992*; *Mascola et al., 1999*; *Mascola et al., 2000*; *Shibata et al., 1999*; *Baba et al., 2000*; *Parren et al., 2001*; *Hessell et al., 2009a*; *Hessell et al., 2009b*; *Hessell et al., 2010*; *Moldt et al., 2012*), suggests that a vaccine that elicits such antibodies would be protective.

However, with the exception of experiments performed in genetically engineered mice (*Dosenovic et al., 2015*; *Escolano et al., 2016*; *Saunders et al., 2019*), all efforts to induce anti HIV-1 bNAbs by vaccination have produced only sporadic or less than optimal antibody responses with little or no protective activity against heterologous viral strains (*Xu et al., 2018*; *Saunders et al., 2017*). More importantly, it remains unclear which animal model is most relevant to test candidate vaccines.

Because macaque CD4$^+$ T cells are resistant to HIV-1, SIV/HIV chimeric viruses (SHIVs), carrying the HIV-1 Env, were constructed to target and replicate efficiently in monkey CD4$^+$ T lymphocytes in vivo. An intrinsic property of HIV-1 is its replicative vigor and genetic variability in infected persons, both of which confer the capacity to escape immunologic and pharmacologic pressure. Unfortunately, most currently available SHIV challenge viruses do not possess this phenotype and are unable to resist immunologic or cART pressure. This deficiency results in suppressed SHIV replication in many of the inoculated animals and their failure to develop clinical immunodeficiency. In contrast, molecularly cloned SHIV$_{AD8-EO}$ possesses many of the properties intrinsic to HIV-1. It is R5 tropic, produces sustained levels of plasma viremia in inoculated macaques, exhibits a Tier two neutralization sensitivity phenotype, generates resistant variants in bNAb and ART-treated animals, and causes irreversible depletions of CD4$^+$ T cells in infected monkeys (*Shingai et al., 2013*; *Gautam et al., 2012*; *Nishimura et al., 2010*; *Shingai et al., 2012*). SHIV$_{AD8-EO}$ inoculation of macaques invariably leads to symptomatic immunodeficiency associated with opportunistic infections and a fatal clinical outcome in untreated animals. As is the case for some HIV-1 infected individuals, a few SHIV$_{AD8-EO}$ infected monkeys develop neutralizing antibodies against some heterologous clade A, B, and C HIV-1 strains (*Walker et al., 2011b*), including one macaque (CE8J) that exhibited the phenotype observed for HIV-1-infected human elite neutralizers (*Walker et al., 2011b*). The capacity to generate broadly neutralizing activity persisted throughout the 2 year infection of animal CE8J. This monkey ultimately succumbed to immunodeficiency, and was euthanized 117 weeks post-infection. Plasma mapping studies revealed that neutralizing activity of CE8J macaque exclusively targeted the glycan patch associated with the variable 3 (V3) loop on HIV-1 Env, which is in agreement with studies of emerging viral variants in this macaque that presented substitutions of critical amino acids in the V3 region (*Sadjadpour et al., 2013*).

Here, we report on the cloning and molecular characterization of a V3-glycan bNAb produced in this elite neutralizing non-human primate, its structure bound to an HIV-1 Env trimer, and the implications for development of vaccines targeting the V3-glycan patch.

## Results

### Isolation of single Env-specific B cells from a SHIV$_{AD8}$-infected macaque

To isolate bNAbs from macaque CE8J, we purified germinal center (GC) B cells that bound to YU2 gp140 fold-on trimer (YU2) and BG505 SOSIP.664 trimer (BG505), but not to a control antigen, from lymph nodes collected at the time of necropsy, 115 weeks post-SHIV$_{AD8}$ infection (*Figure 1A* and *Figure 1—figure supplement 1A*).

Immunoglobulin heavy chain (IgH) and light chains Lambda (IgL) and Kappa (IgK)-encoding mRNAs were amplified from the isolated Env-specific B cells by PCR using a set of primers

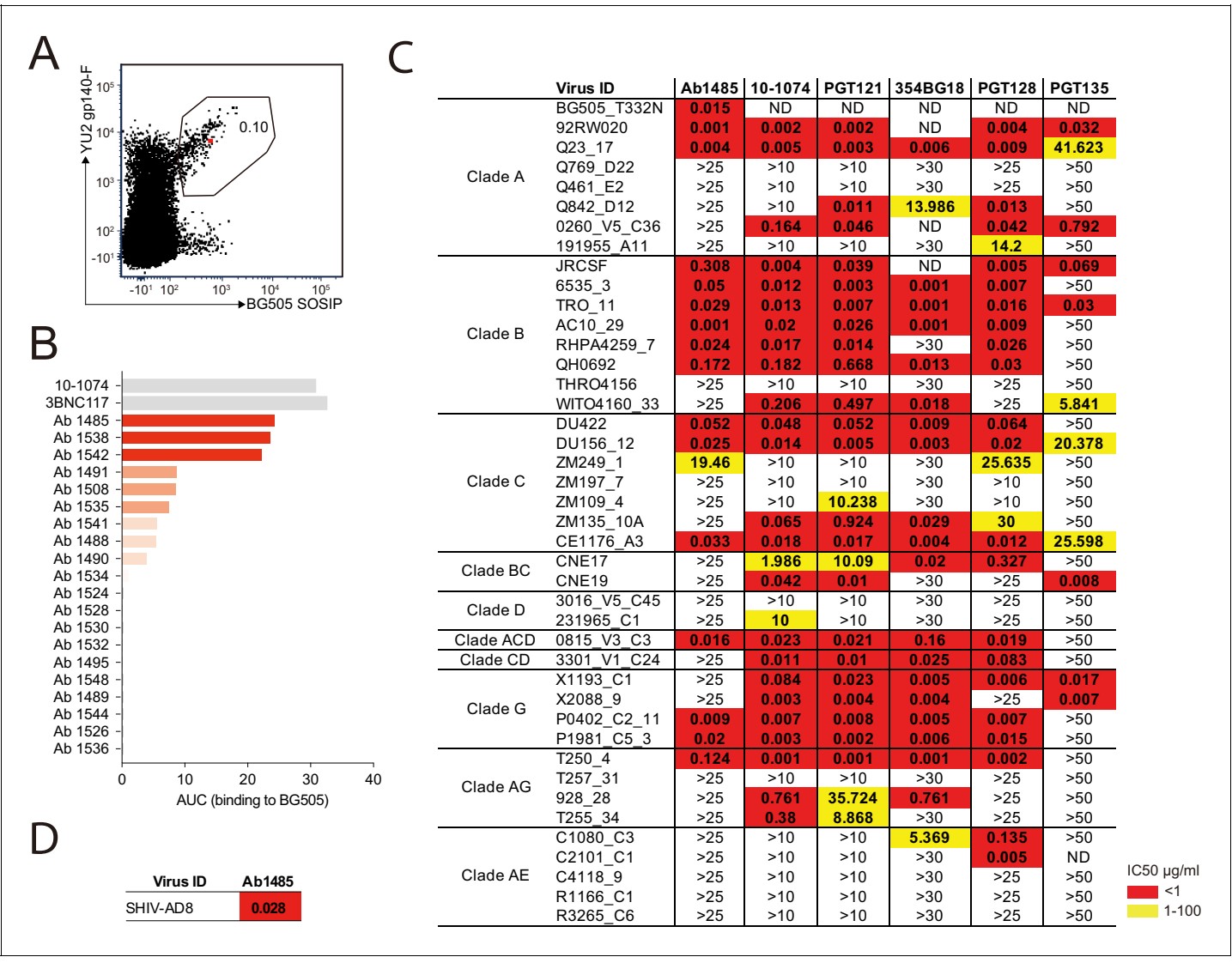

**Figure 1.** Broadly neutralizing antibody isolated from a SHIV$_{AD8}$-infected rhesus macaque. (**A**) FACS plot showing germinal center B cells that bind to YU2 gp140-F and BG505 SOSIP from a lymph node collected from macaque CE8J at week 115 after SHIV$_{AD8}$ infection. The gate shows the sorting window. The B cell carrying the isolated bNAb (Ab1485) is highlighted in red. (**B**) Graph shows the binding of several monoclonal antibodies isolated from macaque CE8J to BG505 SOSIP (Antibodies were tested for binding to BG505 two or three times. Graph shows data from a representative ELISA). Data is shown as area under the ELISA curve (AUC). (**C** and **D**) Table shows the neutralization activity of Ab1485 and other human V3-glycan bNAbs determined in TZM-bl assays against a panel of 42 multi clade tier 1B and tier two pseudoviruses (**C**) and replication-competent SHIV$_{AD8}$ (**D**).

The online version of this article includes the following source data and figure supplement(s) for figure 1:

**Source data 1.** Macaque antibody sequences.

**Figure supplement 1.** Antibodies isolated from a SHIV$_{AD8}$-infected rhesus macaque.

specifically designed to amplify a diverse set of macaque genes (*Escolano et al., 2019*). Paired IgH and IgL/IgK sequences were obtained for 90 antibodies. Among the 90 antibodies, we found two expanded B cell clones (*Figure 1—figure supplement 1B*). Sequence analysis revealed that IgH, IgL and IgK genes were somatically hypermutated (averages of 11.8, 8.5, and 4.1 average nucleotide mutations, respectively) (*Figure 1—figure supplement 1C*). The average length of the complementarity determining region 3 of the heavy chain (CDRH3) was 15.3 amino acids, and 30 of the antibodies had CDRH3s of 18 or more amino acids (*Figure 1—figure supplement 1D*).

## Ab1485 isolated from macaque CE8J is potently neutralizing

We produced 67 of the 90 monoclonal antibodies and tested them for binding in ELISA to BG505, one of the Env baits that we used to isolate Env-binding B cells. Several antibodies showed detectable binding to BG505, with Ab1485 showing the highest-level binding (*Figure 1B*). BG505.664 has previously been reported to bind to human bNAbs and show reduced binding to non-neutralizing antibodies (*Sanders et al., 2013*), thus the three best binders to BG505 were evaluated for neutralizing activity in TZM-bl assays (*Montefiori, 2005*) against a screening panel of 7 HIV-1 pseudoviruses. Only Ab1485 showed potent and broad activity against this panel (*Figure 1—figure supplement 1E*) that could explain the activity previously detected in the plasma of macaque CE8J (*Walker et al., 2011b*). To further evaluate the neutralization potency and breadth of Ab1485, we tested it in TZM-bl assays against a panel of 42 pseudoviruses covering nine different HIV-1 clades and compared its activity to the activity reported for human V3-glycan bNAbs (*Figure 1C*). Ab1485 neutralized 16 of the isolates in the 42-virus panel with a mean $IC_{50}$ of 0.055 µg/mL (*Figure 1C*), and it was also a potent neutralizer of a replication-competent SHIV ($SHIV_{AD8}$, $IC_{50}$ = 0.028 µg/mL, *Figure 1D*). Ab1485 showed broader activity against this panel of pseudoviruses than the human V3-glycan bNAb PGT135, which neutralizes seven isolates with a mean $IC_{50}$ of 0.14 µg/mL, but lower breadth than 10–1074 (25 isolates with a mean $IC_{50}$ of 0.16 µg/mL), PGT121 (23 isolates with a mean $IC_{50}$ of 0.1 µg/mL), 354BG18 (19 isolates with a mean $IC_{50}$ of 0.06 µg/mL) and PGT128 (22 isolates with a mean $IC_{50}$ of 0.04 µg/mL). The mean $IC_{50}$ of Ab1485 against the 14 pseudoviruses for which there is available neutralization data for all these human bNAbs is identical to the mean $IC_{50}$ of PGT121, 0.06 µg/mL, and significantly lower than the mean $IC_{50}$ of PGT13, 17.5 µg/mL. We conclude that Ab1485 is a potent neutralizer with limited breadth compared to some of the human bNAbs reported to date (*Sok and Burton, 2018*).

## Mapping the Ab1485 epitope on HIV-1 Env

Ab1485 combines the germline V-gene segments VH4_2M and VL124_30 and is highly mutated (33 and 25 nucleotide mutations in the VH and VL genes respectively). It has a 20-residue CDRH3 (*Figure 2A*). We did not find any other family members of Ab1485 when analyzing the LN biopsy collected from macaque CE8J at the time of necropsy (week 115 after $SHIV_{AD8}$ challenge). In an attempt to find other clonal relatives of Ab1485, and in the absence of available LN biopsies at earlier time points after $SHIV_{AD8}$, we aimed to isolate Env-specific B cells from a PBMC sample collected at week 38 after viral infection using RC1 and BG505 as baits, however, this approach did not yield any B cells to analyze (*Figure 1—figure supplement 1F*). To characterize the target epitope of Ab1485, we performed competition ELISAs using the bNAbs 10–1074, 3BNC117, 8ANC195, PG9, VRC34 that target the V3-glycan patch, the CD4-binding site, the gp120-gp41 interface, the apex and the fusion peptide of Env, respectively (*Mouquet et al., 2012*; *Scheid et al., 2011*; *Walker et al., 2009*; *Kong et al., 2016*). Binding of these antibodies was self-inhibitory and in addition, 3BNC117 was partially inhibited by the gp120-gp41 interface bNAb 8ANC195 (*Scheid et al., 2011*; *Scharf et al., 2014*; *Scharf et al., 2015*), and vice versa. The binding of Ab1485 to BG505 was inhibited by the V3-glycan bNAb 10–1074 but not by any of the other human bNAbs (*Figure 2B*).

To further map the antibody target site, we performed neutralization assays using a series of HIV-$1_{JRCSF}$ mutants (*Escolano et al., 2019*). The neutralizing activity of Ab1485 was dependent on the potential N-linked glycosylation sites at $N332_{gp120}$ and $N156_{gp120}$, but unaffected by mutations that interfere with the binding of human bNAbs to the interface (N611D), the CD4-binding site (T278A+ A281T), the V1V2 apex (N160K), the MPER (F673L) or the fusion peptide (N88Q) (*Figure 2C*). In

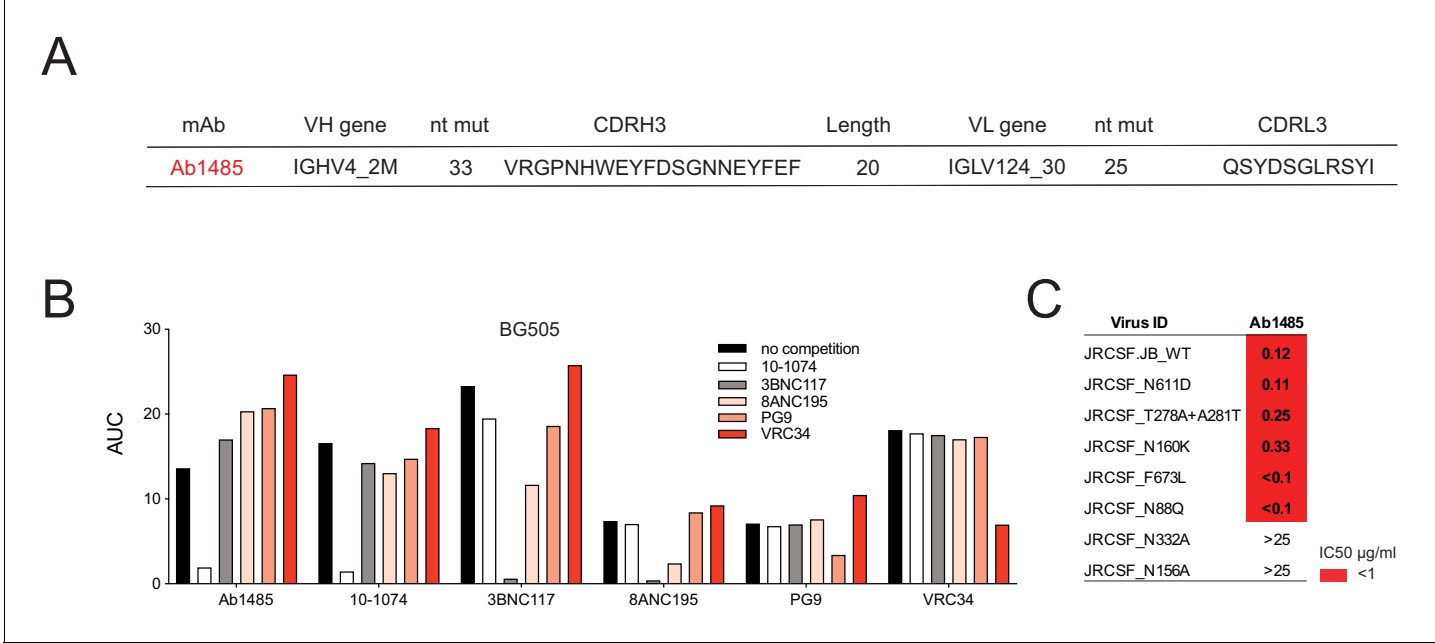

**Figure 2.** Mapping of Ab1485 binding to Env. (**A**) Description of Ab1485. (**B**) Representative ELISA graph showing binding of Ab1485 to BG505 in competition with antibodies that target the V3-glycan epitope (10–1074), the CD4- binding site (3BNC117), the gp120-gp41 interface (8ANC195), the apex (PG9) or the fusion peptide (VRC34) and in the absence of a competing antibody (n = 3). (**C**) Table shows the neutralization activity of Ab1485 determined in TZM-bl assays against a JRCSF pseudovirus and a series of JRCSF mutants that affect the binding of human bNAbs to the interface (N611D), the CD4-binding site (T278A+ A281T), the apex (N160K), the MPER (F673L), and the fusion peptide (N88Q).

conclusion, the competition ELISA and neutralization results suggested that Ab1485 targets the V3-glycan patch in Env.

## Cryo-EM structure of Ab1485 in complex with BG505

To elucidate the molecular details of Env recognition by Ab1485, we determined a 3.5 Å cryo-EM structure of Ab1485 in complex with the BG505 SOSIP.664 trimer and the gp120-gp41 interface targeting antibody, 8ANC195 (*Scharf et al., 2014*; *Scharf et al., 2015*; *Figure 3A*, *Figure 1—source data 1*-table 1 and *Figure 3—figure supplement 1*). The structure of the Ab1485-Env complex revealed recognition of an epitope focused on the $N332_{gp120}$ glycan, the gp120 GDIR peptide motif, and V1 loop (*Figure 3B*). In common with human-derived V3/$N332_{gp120}$ glycan-targeting bNAbs, Ab1485's primary interaction was with the $N332_{gp120}$ glycan ($Man_6GlcNAc_2$), which interfaces almost entirely with the CDRH3 loop (*Figure 3C*) (~400 Å$^2$ antibody buried surface area (BSA)). In addition to the $N332_{gp120}$ glycan, Ab1485 makes secondary contacts with the $N156_{gp120}$ glycan (~275 Å$^2$ antibody BSA), which frames Ab1485's VH domain at the gp120 V3 epitope, rationalizing the observed loss in neutralization activity when tested against the JRCSF $\Delta N156_{gp120}$ glycan virus (*Figure 2C*) and consistent with faster dissociation kinetics observed in surface plasmon resonance (SPR) binding experiments that demonstrated a faster dissociation rate for a SOSIP that lacks the $N156_{gp120}$ glycan (RC1) compared to a SOSIP that contains the $N156_{gp120}$ potential N-linked glycosylation site (BG505) (*Figure 3—figure supplement 2A–C*).

Despite sharing a common interaction with the $N332_{gp120}$ glycan, numerous studies have demonstrated that the V3-targeting bNAbs can adopt different binding orientations when targeting the $N332_{gp120}$ glycan supersite (*Kong et al., 2013*; *Freund et al., 2017*; *Pancera et al., 2014*). When compared to the poses of human-derived bNAbs (*Figure 3D–F*) and V3-targeting antibodies elicited in rabbits or rhesus macaques by vaccination or SHIV$_{BG505}$ challenge (*Figure 3—figure supplement 2D*), Ab1485 adopts a Env-binding orientation in a manner most closely related to PGT128, which primarily uses its heavy chain to interact with the V3/$N332_{gp120}$-glycan epitope (*Kong et al., 2015*; *Figure 3D–F*). However, in contrast to PGT128, Ab1485 lacks any V-gene insertions or deletions and

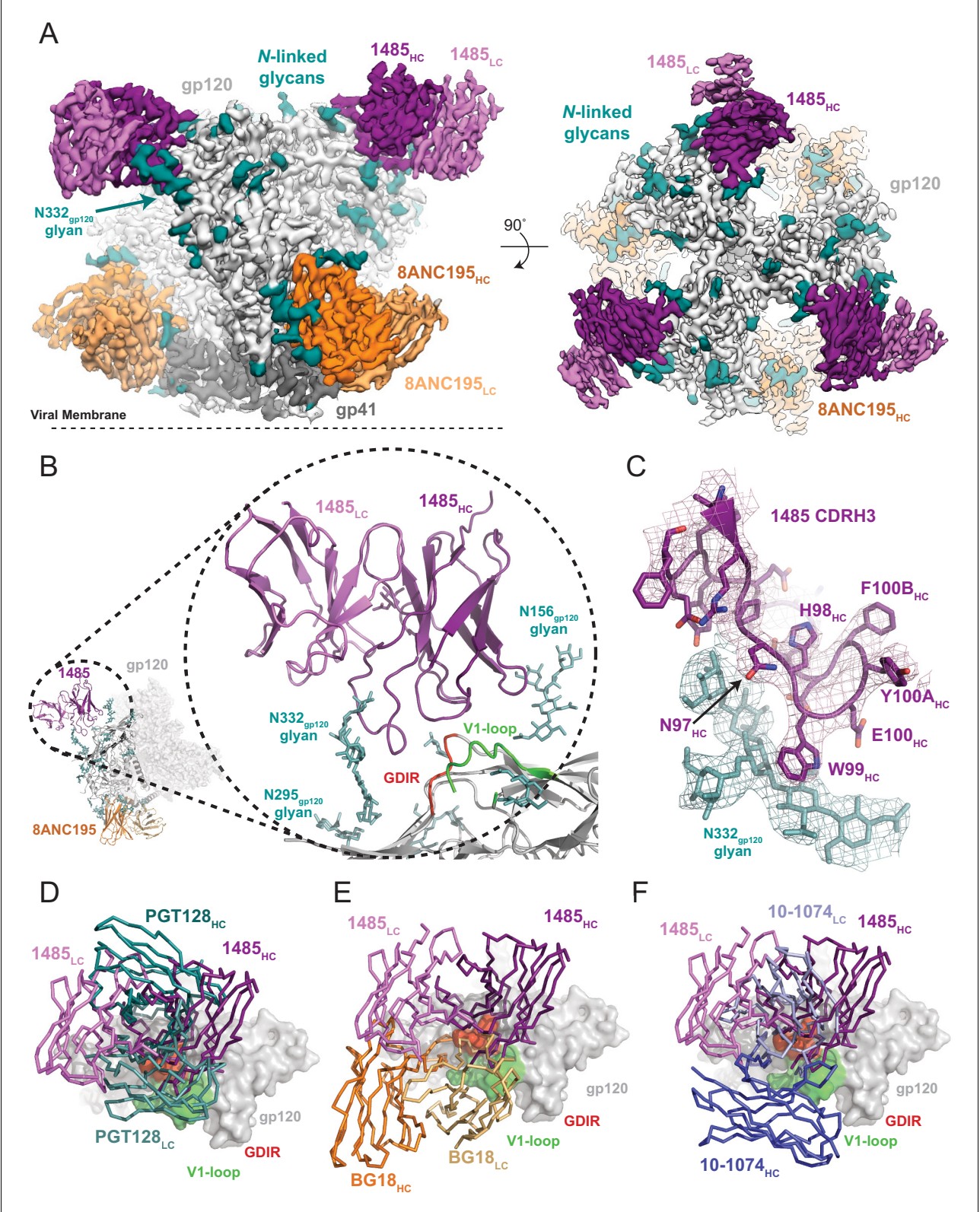

**Figure 3.** Cryo-EM reconstruction of the Ab1485-BG505 complex reveals a distinct Env-binding orientation relative to human bNAbs. (**A**) Cryo-EM map of the BG505 SOSIP.664 trimer bound to three Ab1485 (purple shades) and three 8ANC195 (orange shades) Fabs. Densities for glycans are colored in dark teal. (**B**) Cartoon depiction of the modeled complex with a close-up view of the Ab1485 Fab – gp120 interface. Conserved regions of the V3-epitope are highlighted. (**C**) Cartoon and stick representation of the Ab1485 CDRH3 recognition of the N332$_{gp120}$-glycan. Reconstructed cryo-EM map

*Figure 3 continued on next page*

*Figure 3 continued*

shown as a mesh, contoured at three sigma. (**D–F**) Comparison of Ab1485's (purple) Env-binding orientation to (**D**) PGT128 (teal, PDB 5ACO), (**E**) BG18 (orange, PDB 6CH7), and (**F**) 10–1074 (blue, PDB 5T3Z).

The online version of this article includes the following figure supplement(s) for figure 3:

**Figure supplement 1.** Cryo-EM data processing and validation.

**Figure supplement 2.** Binding assays and structural comparison to V1//V3 antibodies raised in animals.

its orientation is rotated ~90° relative to PGT128, resulting in a unique Env-binding orientation that shifts interactions away from the $N301_{gp120}$ N-glycan and towards the $N156_{gp120}$ glycan, and moves the light chain outside of the underlying V3-epitope (*Figure 3* and *Figure 4A–C*). Thus, interactions with the N-linked glycans and gp120 peptide components are almost exclusively mediated by the Ab1485 heavy chain (*Figure 4A,B*; 1255 $Å^2$ vs 65 $Å^2$ buried surface areas for HC and LC components of Ab1485 paratope, respectively). This observation suggests that Ab1485 may not be restricted by light chain pairing or require the consensus light chain sequence motifs commonly observed in human-derived V3/$N332_{gp120}$-glycan-targeting bNAbs.

Closer examination of the Ab1485 epitope revealed that all three heavy chain CDR loops contact the $_{324}GDIR_{327}$ gp120 peptide motif at the base of the V3-loop, which contrasts interactions by both heavy and light chain CDR loops observed in 10–1074/PGT121-like and BG18 bNAbs (*Gristick et al., 2016*; *Barnes et al., 2018*; *Figure 4B,D*). The primary molecular contacts are with CDRH1 residues $R30_{HC}$, $S31_{HC}$, and $N32_{HC}$, which form potential hydrogen bonds with backbone and sidechain atoms from residues $G324_{gp120}$ and $D325_{gp120}$ (*Figure 4E*). Interestingly, these CDRH1 residues mimic interactions made by CDRL1/L3 residues in PGT121/101074-like bNAbs with Env, providing evidence for a convergent chemical mechanism of interactions with the gp120 GDIR motif, as previously shown for BG18 (*Barnes et al., 2018*). Moreover, Ab1485 residue $R30_{HC}$ forms secondary interactions with V1 loop residues $V134_{gp120}$ and $T135_{gp120}$, resembling similar arginine-gp120 V1-loop interactions observed in 10–1074, BG18, and PGT128 (*Kong et al., 2015*; *Gristick et al., 2016*; *Barnes et al., 2018*).

A common interaction seen in the PGT121/101074-like, BG18, and PGT128 bNAbs involves the formation of a salt bridge between $R327_{gp120}$ and an either a glutamate or aspartate in CDRH3 (12, 58). The Ab1485-BG505 structure reveals an alternate binding mode that involves the guanidinium moiety of $R327_{gp120}$ forming cation-pi stacking interactions with $W99_{HC}$ at the tip of the Ab1485's CDRH3, while also participating in hydrogen bonding with the backbone carbonyl group (*Figure 4E*). While CDRH3 residue $E100_{HC}$ could potentially adopt a conformation that would promote salt-bridge formation with $R327_{gp120}$, this residue is pointed outward and within H-bonding distance to the $N332_{gp120}$-glycan in the Ab1485-BG505 complex. This observation suggests that salt-bridge formation between $R327_{gp120}$ and an acidic CDRH3 residue found in V3-targeting bNAbs may not be as critical to targeting the V3/N332gp120-glycan epitope.

## Ab1485 protects from infection by SHIV$_{AD8}$ in rhesus macaques

To determine whether Ab1485 can protect against SHIV$_{AD8}$ infection in macaques, we expressed a fully macaque IgG, including Fc region substitutions that increase half-life through rescue by increased FcRn recycling (Ab1485-LS) (*Zalevsky et al., 2010*). Ab1485 was not polyreactive, as shown by ELISAs against a series of antigens (*Figure 5—figure supplement 1A*) and Ab1485-LS had a half-life of 2.67 days in transgenic human FcRn mice (*Roopenian et al., 2003*; *Figure 5—figure supplement 1B*). The protective efficacy of Ab1485-macaque-LS was assessed in rhesus macaques that received a single high dose of SHIV$_{AD8}$ intrarectally (1000 TCID50) one day after a single intravenous infusion of Ab1485 at 10 mg/kg body weight (*Figure 5A*). Two historical control monkeys (*Yamamoto, 2015*) receiving no mAb, rapidly became infected, generating peak levels of plasma viremia on day 14 post-challenge. In contrast, the four macaques receiving Ab1485 remained uninfected throughout a 25-week observation period (*Figure 5B*). Neutralizing antibody titers persisted in the peripheral blood for at least 50 days after the virus challenge (*Figure 5C*). We conclude that Ab1485-macaque-LS protects macaques from high dose intrarectal SHIV$_{AD8}$ infection.

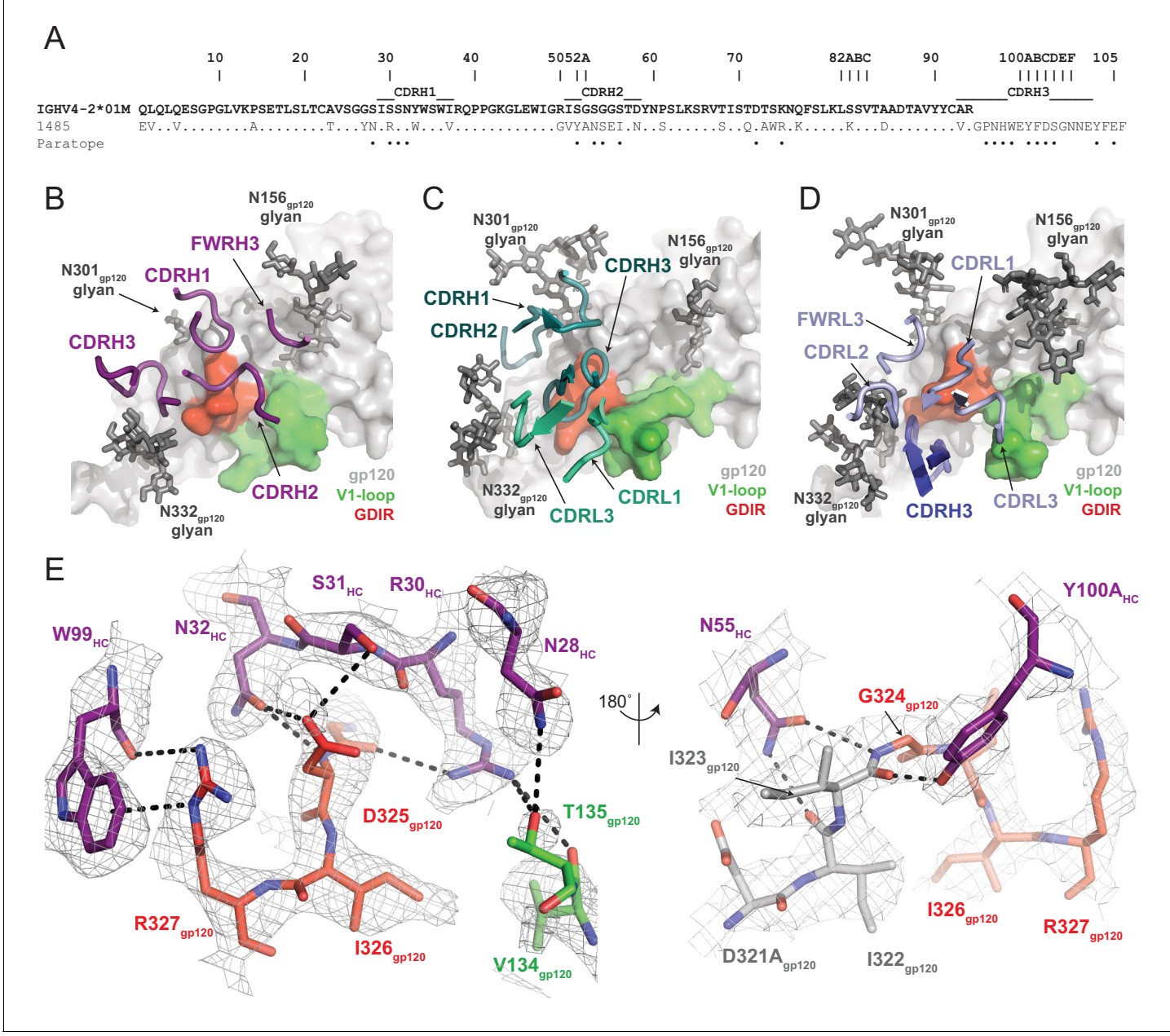

**Figure 4.** Molecular details of Ab1485-gp120 interactions. (A) Sequence alignment of mature Ab1485 heavy chain with germline VH4-2*01M. Paratope residues are highlighted. (B–D) Comparison of the paratope CDR loops and FWRs involved in epitope recognition for (B) Ab1485, (C) PGT128, and (D) 10–1074. (E) Stick representation of interactions between Ab1485 (purple) and either the GDIR peptide motif (red) or V1 loop (green). Potential H-bonds defined are shown as black dashes.

## Discussion

Indian-origin rhesus macaques infected with SHIV are an important model for evaluating HIV-1 prevention and therapy strategies. Macaques can also potentially be used to evaluate humoral immune responses to candidate HIV-1 vaccines, but whether macaques produce human-like bNAbs has not been evaluated. We have examined the antibody response of macaque CE8J who developed broad and potent serologic activity against HIV-1 many months after infection with SHIV_AD8. Monoclonal

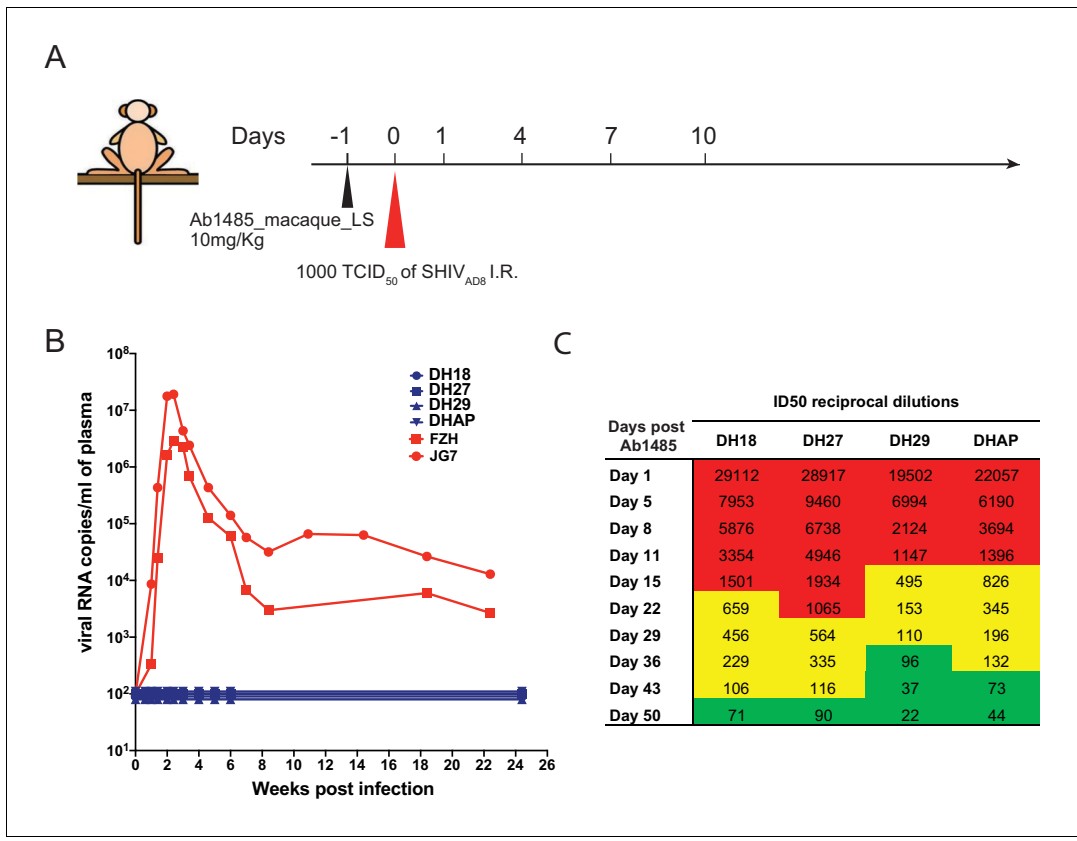

**Figure 5.** Ab1485 protects macaques against a high dose intrarectal challenge with SHIV_AD8. (**A**) Diagrammatic representation of the regimen used to assess the protective efficacy of Ab1485. Macaques were administered with Ab1485 at a dose of 10 mg kg$^{-1}$ and challenged one day later with 1000 TCID$_{50}$ of SHIV$_{AD8}$ intrarectally (I.R.) (**B**) Longitudinal analysis of plasma viral loads in two control macaques (FZH and JG7) receiving no Ab and four macaques (DH18, DH27, DH29, and DHAP) infused with Ab1485 24 hr prior to SHIV$_{AD8}$ challenge. (**C**) Serum neutralizing antibody titers in macaques receiving Ab1485. The IC$_{50}$ titers are color coded: 1:21–99 in green; 1:100–999 in yellow and ≥1:1000 in red.

The online version of this article includes the following figure supplement(s) for figure 5:

**Figure supplement 1.** Characterization of Ab1485.

Ab1485 obtained from single-B-cells purified on the basis of HIV-1 Env-binding by cell sorting neutralized 16 of 42 of the HIV-1 strains tested with IC$_{50}$ lower than 1 µg.ml. Through biochemical and structural analysis, we determined that antibody 1485 targets the V3/N332$_{gp120}$-glycan epitope and does so in a manner that is similar to the human V3-targeting antibodies. Thus, rhesus macaques develop anti-HIV-1 V3-glycan patch bNAbs that resemble human bNAbs.

As many as 10–20% of HIV-1 infected humans develop antibodies that can neutralize a number of different HIV-1 strains, but only an elite few (1–2%) produce potent broadly neutralizing serologic activity (*Landais and Moore, 2018*). The elite humoral responders typically take 1–3 years to produce bNAbs, which is highly unusual for an antibody response. Single cell antibody cloning revealed that human bNAbs carry large numbers of somatic mutations that are required for their neutralizing activity (*Mouquet et al., 2012*; *Scheid et al., 2011*; *Scheid et al., 2009*; *Xiao et al., 2009*; *Muster et al., 1993*; *Zhou et al., 2010*; *Hoot et al., 2013*; *Klein et al., 2013*; *Sok et al., 2013*; *Kepler et al., 2014*; *Bonsignori et al., 2011*). This observation led to the proposal that the development of bNAbs required a prolonged series of sequential interactions between the antibody-producing B lymphocytes and virus escape variants to drive antibody maturation (*Scheid et al., 2009*; *Dimitrov, 2010*). Elegant prospective studies of virus and antibody evolution supported this concept

(*Liao et al., 2013*; *Bonsignori et al., 2017*; *Doria-Rose et al., 2014*; *Moore et al., 2012*; *Bhiman et al., 2015*), and sequential immunization experiments reproduced it in knock in mice (*Escolano et al., 2016*). Monoclonal antibody Ab1485 resembles human bNAbs in many important respects including the high levels of somatic mutation suggesting a requirement for sequential interaction between B cells and the virus to drive bNAb evolution in macaques. This process of virus and antibody co-evolution has been recently shown in macaques in a study by Roark and colleagues (preprint in bioRxiv, *Roark et al., 2020*). Their work shows that infection with SHIVs bearing the Envs of transmitted founder (T/F) HIV-1 viruses that induced bNAbs in humans, recapitulates key events of the previously reported interplay between HIV-1 and antibodies in the infected individuals. Infection with one of these T/F-SHIVs, SHIV.CH505, induced the development of a lineage of V2-apex bNAbs that shows certain breadth and potency against tier 1 and tier two pseudoviruses. The most mutated member of this lineage, RHA1.V2.01, has comparable frequencies of IgH V(D)J nucleotide mutations to Ab1485 (8.5% in RHA1.V2.01 *vs* 8.6% in Ab1485) and slightly lower mutation frequencies in the IgL (6.2% in RHA1.V2.01 *vs* 7.55% in Ab1485). Regarding the neutralization activity, RHA1.V2.01 and Ab1485 show similar breadth when considering isolates neutralized at $IC_{50}$ <1 µg/mL (30.7% for RHA1.V2.01 vs 38.1% for Ab1485) but Ab1485 is more potent against the reported neutralized isolates (mean $IC_{50}$ of RHA1.V2.01 = 0.214 µg/mL *vs* mean $IC_{50}$ of Ab1485 0.055 µg/mL).

Monoclonal antibody Ab1485 targets the glycan patch that surrounds the conserved GDIR peptide at the base of the V3 loop, an epitope that is frequently targeted by human bNAbs. Ab1485 resembles PGT128 in that most of the interactions with the V3/$N332_{gp120}$-glycan epitope are mediated by heavy chain CDR loops, including conserved interactions with the $N332_{gp120}$ glycan and gp120 GDIR peptide motif. Recent studies in macaques showed that after priming with a designed immunogen that focuses the response to the V3-glycan epitope, antibodies with binding features of V3-glycan-targeting bNAbs can be isolated (*Escolano et al., 2019*). Whether these early antibodies can mature to develop broad neutralizing activity remains to be determined, but antibodies like Ab1485 may provide a blueprint for achieving such broad and potent activity.

Some of the predicted impediments for Env-binding of germline forms of PGT121/10-1074-like or BG18-like antibodies are related to overcoming unfavorable interactions with the antibody light chain, particularly with the gp120 V1 loop. The unique orientation adopted by Ab1485, which positions the LC away from the V1 loop, may provide an easier path towards antibody maturation. Overall, structural analysis of 1485 provides: (i) evidence that effective bNAbs targeting the V3/$N332_{gp120}$-glycan epitope are not restricted to PGT128, BG18 or 10–1074/PGT121-like binding mechanisms, (ii) critical structural insights towards immunogen design efforts to elicit neutralizing antibodies, including alternative binding modes that do not include light chain interactions, and (iii) evidence that $SHIV_{AD8}$-infected macaques are capable of generating bNAbs that target the V3/$N332_{gp120}$-glycan epitope in a manner similar to human-derived bNAbs, and therefore represent an excellent animal model for developing HIV-1 vaccines targeting this site.

Human bNAbs can protect humanized mice and macaques from HIV-1 and SHIV infections, respectively, when given prophylactically (*Eichberg et al., 1992*; *Emini et al., 1992*; *Mascola et al., 1999*; *Mascola et al., 2000*; *Shibata et al., 1999*; *Baba et al., 2000*; *Parren et al., 2001*; *Hessell et al., 2009a*; *Hessell et al., 2009b*; *Hessell et al., 2010*; *Moldt et al., 2012*; *Pietzsch et al., 2012*; *Gruell et al., 2013*). They can also suppress infection for prolonged periods of time in mice and macaques (*Shingai et al., 2013*; *Barouch et al., 2013*; *Nishimura et al., 2017*; *Horwitz et al., 2013*). When administered to Indian-origin rhesus macaques during the acute $SHIV_{AD8}$ infection they induced host CD8+ T cell-dependent immunity that can suppress infection for 2 to 3 years (*Nishimura et al., 2017*). However, prolonged administration of these human monoclonal antibodies to macaques has not been possible due to rapid development of macaque anti-human antibody responses. The fully-native macaque Ab1485 will facilitate such therapies and reservoir reduction experiments that require prolonged bNAb administration to macaques.

Finally, the observation that Indian-origin rhesus macaques develop V3-glycan patch bNAbs that resemble human bNAbs is strongly supportive of the use of this model organism to test HIV-1 candidate vaccines that target this epitope.

# Materials and methods

## Key resources table

| Reagent type (species) or resource | Designation | Source or reference | Identifiers | Additional information |
|---|---|---|---|---|
| strain, strain background (*Mus musculus* Female) | B6. Cg-Fcgrt^tm1Dcr Tg(CAG-FCGRT)276Dcr/DcrJ mice (FcRn -/- hFcRn) | The Jackson Laboratory | 004919 | 7–8 weeks of age |
| cell line (human) | HEK293-6E | National Research Council of Canada | RRID:CVCL_HF20 | |
| Cell line (human) | Expi293F cells | Invitrogen | A14527 | |
| cell line (CHO) | CHO Flp-InTM cells | Invitrogen | R75807 | *Chung et al., 2014* PMID:24767177 |
| Recombinant protein | Avi-tagged BG505 SOSIP | Kindly provided by Dr. Rogier W. Sanders and Dr. Marit van Gils. | | *Sok et al., 2013* PMID:25422458 |
| Recombinant protein | Avi-tagged YU2-gp140 fold-on | Kindly provided by Dr. R. Wyatt | | |
| Recombinant protein | Avi-tagged Hepatitis B surface antigen | Protein Specialists | hbs-875 | |
| Recombinant protein | Avi-tagged RC1 | | | *Escolano et al., 2019* PMID:31142836 |
| Recombinant protein | Avi-tagged RC1_KO | | | *Escolano et al., 2019* PMID:31142836 |
| Recombinant protein | BG505 SOSIP.664 trimer | | | *Schoofs et al., 2019* PMID:31126879 |
| Commercial assay or kit | Zombie NIR Fixable Viability Kit | Biolegend | 77184 | (1:400) |
| Commercial assay or kit | TCL lysis buffer | Qiagen | 1031576 | |
| Commercial assay or kit | RNAClean XP | Beckman Counter | A63987 | |
| antibody | anti CD3-APC-eFluor 780 (mouse monoclonal) | Invitrogen | 47-0037-41 | (1:200) |
| antibody | anti CD14-APC-eFluor 780 (mouse monoclonal) | Invitrogen | 47-0149-42 | (1:200) |
| antibody | anti CD38 FITC | Stem Cell | 60131FI | (1:200) |
| antibody | anti CD16-APC-eFluor 780 (mouse monoclonal) | Invitrogen | 47-0168-41 | (1:200) |
| antibody | anti CD8-APC-eFluor 780 (mouse monoclonal) | Invitrogen | 47-0086-42 | (1:200) |

*Continued on next page*

*Continued*

| Reagent type (species) or resource | Designation | Source or reference | Identifiers | Additional information |
|---|---|---|---|---|
| antibody | anti CD20-PE-Cy7 (mouse monoclonal) | BD Biosciences | 335793 | (1:200) |
| antibody | HRP-conjugated anti-human IgG (Fc) CH2 Domain antibody (mouse monoclonal) | Bio-Rad | MCA647P | (1:5000) |
| software, algorithm | WinNonlin 6.3 | Certara Software | | |
| software, algorithm | MacVector v.17.0.2 | MacVector | | |
| software, algorithm | FlowJo v.10.6.1 | Becton Dickinson | | |
| software, algorithm | FCS EXPRESS | De Novo | | |
| software, algorithm | GraphPad Prism 7 | GraphPad | | |
| software, algorithm | SeqIO | Biopython | | |
| software, algorithm | cutadapt v.2.3 | cutadapt | | |
| Software, algorithm | Change-O toolkit v.0.4.5 | | | PMID:26069265 |
| software, algorithm | SerialEM v3.7 | | RRID:SCR_017293 | *Mastronarde, 2005* PMID:16182563 |
| software, algorithm | cryoSPARCv2.14 | | RRID:SCR_016501 | *Punjani et al., 2017* PMID:28165473 |
| software, algorithm | Relion v3.0 | | RRID:SCR_016274 | *Zivanov et al., 2018* PMID:30412051 |
| software, algorithm | UCSF Chimera v1.13 | | RRID:SCR_004097 | *Goddard et al., 2007*, PMID:16963278 |
| software, algorithm | Phenix v1.17 | | RRID:SCR_014224 | Afonine, et al, ACTA D, 2018, PMID:29872004 |
| software, algorithm | Coot v0.8.9 | | RRID:SCR_014222 | *Emsley et al., 2010*, PMID:20383002 |
| software, algorithm | PyMOL v1.8.2.1 | Schrodinger, Inc | RRID:SCR_000305 | https://pymol.org/2/ |
| software, algorithm | Biacore T200 Evaluation Software v3.2 | Cytiva | | |

## Flow cytometry and single-B-cell sorting

Frozen lymph node (LN) cell suspensions collected from macaque CE8J at 117 weeks post-infection were thawed and incubated in FACS buffer (1 X Phosphate-buffered saline (PBS), 2% calf serum, 1 mM EDTA.) with the following anti human antibodies: anti CD3-APC-eFluor 780 (Invitrogen, 47-0037-41), anti CD14-APC-eFluor 780 (Invitrogen, 47-0149-42), anti CD16-APC-eFluor 780 (Invitrogen, 47-0168-41), anti CD8-APC-eFluor 780 (Invitrogen, 47-0086-42), anti CD20-PE-Cy7 (BD biosciences, 335793), and anti CD38 FITC (Stem Cell, 60131FI) and the Zombie NIR Fixable Viability Kit (Biolegend, 77184). Avi-tagged and biotinylated BG505 SOSIP, YU2-gp140 fold-on, and hepatitis B surface antigen (HBs Ag) (Protein Specialists, hbs-875) conjugated to streptavidin Alexa Fluor 647 (Biolegend, 405237), streptavidin PE (BD biosciences, 554061) and streptavidin BV711(BD

biosciences, 563262) respectively were added to the antibody mixture and incubated with the LN cells for 30 min. Single CD3⁻CD8⁻CD14⁻CD16⁻ CD20⁺CD38⁺ BG505 SOSIP⁺YU2-gp140⁺ B cells were sorted into individual wells of a 96-well plates containing 5 µl of a lysis buffer (Qiagen, 1031576) per well using a FACS Aria III (Becton Dickinson). The sorted cells were stored at −80°C or immediately used for subsequent RNA purification (*Escolano et al., 2019*; *Wang et al., 2020*).

A PBMC sample collected from macaque CE8J at week 38 post-SHIVAD8 infection was thawed and incubated in FACS buffer with the following anti human antibodies: anti CD3-APC-eFluor 780 (Invitrogen, 47-0037-41), anti CD14-APC-eFluor 780 (Invitrogen, 47-0149-42), anti CD16-APC-eFluor 780 (Invitrogen, 47-0168-41), anti CD8-APC-eFluor 780 (Invitrogen, 47-0086-42), and anti CD20-PE-Cy7 (BD biosciences, 335793), and the Zombie NIR Fixable Viability Kit (Biolegend, 77184). Avi-tagged and biotinylated RC1_KO, RC1, and BG505 SOSIPs conjugated to streptavidin BV605, streptavidin PE and APC, and streptavidin PerCP Cy5.5 respectively were added to the antibody mixture and incubated with the PBMCs for 30 min.

The use of streptavidin tetramers to facilitate the detection of the interaction between the B cells and Env may allow the isolation of B cells that bind Env with low affinity. The binding avidity under these conditions may facilitate the detection of B cell-Env interactions that are not detected by other methods such as ELISAs.

## Single-B-cell antibody cloning

Single cell RNA was purified from individual B cells using magnetic beads (Beckman Counter, RNA-Clean XP, A63987) and used for cDNA synthesis by reverse transcription (SuperScript III Reverse Transcriptase, Invitrogen, 18080–044, 10,000 U). cDNA was stored at −20°C or used for subsequent amplification of the variable IgH, IgL, and IgK genes by nested PCR and SANGER sequencing using the primers and protocol previously described (*Escolano et al., 2019*).

IgH, IgL, and IgK V(D)J genes were cloned into expression vectors containing the human IgG1, IgL, or IgK constant region using sequence and ligation independent cloning (SLIC) as previously described (*Escolano et al., 2019*; *von Boehmer et al., 2016*). Ab1485-macaque-LS was cloned using pre designed gene fragments (IDT).

The primers used to amplify the immunoglobulin genes anneal at the beginning of the framework region one and the end of the J genes, therefore, this method is likely to introduce point mutations in these regions that may affect the binding properties of the cloned antibodies.

The sorting strategy does not include staining for the antibody isotype, and the primers used to amplify the antibody genes anneal outside the constant region of the antibody genes, therefore, the original isotype of the sorted antibodies is not known. All the isolated antibodies were cloned with an IgG1 constant region, which may affect the binding properties of the cloned antibodies.

## Antibody production

IgGs were expressed by transient transfection in HEK293-6E cells and purified from cell supernatants using protein A or G (GE Healthcare) as previously described (*Escolano et al., 2019*; *Wang et al., 2020*).

## ELISA

ELISAs using BG505 SOSIP directly coated on a 96-well plate (Life Sciences, #9018) were performed as previously described (*Escolano et al., 2019*). Briefly, high affinity 96-well flat bottom plates were coated with the SOSIP at 2 µg/mL and incubated overnight at 4°C. The plates were washed three times with PBS-0.05% Tween 20 and blocked with 2% of milk for 1 hr at RT. After blocking, monoclonal antibodies were added to the plate at 3-fold serial dilutions starting at 10 µg/mL and incubated for 2 hr at RT. Binding was developed with a horseradish peroxidase (HRP)-conjugated anti-human IgG secondary antibody (Jackson ImmunoResearch, 109-035-088) and using ABTS as the HRP substrate.

For competition ELISA, 96-well flat bottom plates were first coated with streptavidin (2 µg/mL) at 37°C for 1 hr, then washed and blocked with 2% milk and incubated with biotinylated BG505 SOSIP (2 µg/mL) at 37°C for 1 hr. Competing bNAb Fabs (10–1074, 3BNC117, 8ANC195, VRC34, and PG9) were added at 10 µg/mL to the plates for 1 hr at 37°C. After wash, serially diluted mAbs were added and incubated at 37°C for 2 hr. The binding was detected by an HRP-conjugated anti-human IgG

(Fc) CH2 Domain antibody (Bio-Rad MCA647P) used at a 1:5000 dilution at RT for 1 hr and developed as described above.

## Polyreactivity assay

ELISAs to determine antibody binding to LPS, KLH, single stranded DNA, dsDNA and insulin were previously described in *Gitlin et al., 2016*. ED38 (*Wardemann et al., 2003*; *Meffre et al., 2004*) and mG053 (*Yurasov et al., 2005*) antibodies were used as positive and negative controls.

## Antibody pharmacokinetic analysis

Female B6. Cg-Fcgrt$^{tm1Dcr}$ Tg(CAG-FCGRT)276Dcr/DcrJ mice (FcRn -/- hFcRn) (The Jackson Laboratory, #004919) aged 7–8 weeks were intravenously injected (Retro-orbital vein) with 0.5 mg of purified Ab1485-macaque-LS in PBS. Total serum concentrations of human IgG were determined by ELISA as previously described with minor modifications (*Klein et al., 2012*). In brief, high-binding ELISA plates (Corning) were coated with Goat Anti-Human IgG (Jackson ImmunoResearch #109-005-098) at a concentration of 2.5 µg/mL overnight at RT. Subsequently, wells were blocked with blocking buffer (2% BSA (SIGMA), 1 mM EDTA (Thermo Fisher), and 0.1% Tween 20 (Thermo Fisher) in PBS). A standard curve was prepared using human IgG1 kappa purified from myeloma plasma (Sigma-Aldrich). Serial dilutions of the IgG standard (in duplicates) and serum samples in PBS were incubated for 60 min at 37C, followed by HRP-conjugated anti-human IgG (Jackson ImmunoResearch #109-035-008) diluted 1: 5000 in blocking buffer for 60 min at RT. Following the addition of TMB (Thermo Fisher #34021) for 8 min and stop solution, optical density at 450 nm was determined using a microplate reader (BMG Labtech). Plates were washed with 0.05% Tween 20 in PBS between each step. The elimination half-life was calculated using pharmacokinetics parameters estimated by performing a non-compartmental analysis (NCA) using WinNonlin 6.3 (Certara Software).

## In vitro neutralization assays

The neutralization activity of monoclonal antibodies was assessed using TZM-bl cells as described previously (*Sarzotti-Kelsoe et al., 2014*). Two or three technical replicates per sample were assayed.

## Protein expression and purification for structural studies

Ab1485 and 8ANC195 Fabs were recombinantly expressed by transiently transfecting Expi293F cells (Invitrogen) with vectors encoding antibody light chain and C-terminal hexahistidine-tagged heavy chain genes. Secreted Fabs were purified from cell supernatants harvested four days post-transfection using Ni$^{2+}$-NTA affinity chromatography (GE Healthcare), followed by size exclusion chromatography (SEC) with a Superdex 16/60 column (GE Healthcare). Purified Fabs were concentrated and stored at 4 °C in storage buffer (20 mM Tris pH 8.0, 120 mM NaCl, 0.02% sodium azide).

A gene encoding soluble BG505 SOSIP.664 v4.1 gp140 trimer (*Sanders et al., 2013*) was stably expressed in Chinese hamster ovary cells (kind gift of John Moore, Weill Cornell Medical College) as described (*Dey et al., 2018*). Secreted Env trimers were isolated using PGT145 immunoaffinity chromatography by covalently coupling PGT145 IgG monomer to an activated-NHS Sepharose column (GE Healthcare) as previously described (*Dey et al., 2018*). Properly folded trimers were eluted with 3M MgCl$_2$ and immediately dialyzed into storage buffer before being subjected to multiple size exclusion chromatography runs with a Superdex200 16/60 column followed by a Superose6 10/300 column (GE Healthcare). Peak fractions verified to be BG505 SOSIP.664 trimers were stored as individual fractions at 4 °C in storage buffer.

## Cryo-EM sample preparation

Complexes of Ab1485-BG505-8ANC195 were assembled by incubating purified Fabs with BG505 trimers at a 3:1 Fab:gp140 protomer ratio overnight at room temperature. Complexes were purified from excess Fab by size exclusion chromatography using a Superose-6 10/300 column (GE Healthcare). Peak fractions corresponding to the Ab1485-BG505-8ANC195 complex were concentrated to 1.5 mg/mL in 20 mM Tris pH 8.0, 100 mM NaCl and deposited onto a 400 mesh, 1.2/1.3 Quantifoil grid (Electron Microscopy Sciences) that had been freshly glow-discharged for 45 s at 20 mA using a PELCO easiGLOW (Ted Pella). Samples were vitrified in 100% liquid ethane using a Mark IV Virtobot

(Thermo Fisher) after blotting for 3 s with Whatman No. one filter paper at 22 °C and 100% humidity.

## Cryo-EM data collection, processing, and model refinement

Movies of the Ab1485-BG505-8ANC195 complex were collected on a Talos Arctica transmission electron microscope (Thermo Fisher) operating at 200 kV using SerialEM automated data collection software (*Mastronarde, 2005*) and equipped with a Gatan K3 Summit direct electron detector. Movies were obtained in counting mode at a nominal magnification of 45,000x (super-resolution 0.435 Å/pixel) using a defocus range of −1.5 to −3.0 μm, with a 3.6 s exposure time at a rate of 13.4 e⁻/pix/s, which resulted in a total dose of 60 e⁻/Å$^2$ over 40 frames.

Cryo-EM data processing was performed as previously described (*Schoofs et al., 2019*). Briefly, movie frame alignment was carried out with MotionCorr2 (*Zheng et al., 2017*) with dose weighting, followed by CTF estimation in GCTF (*Zhang, 2016*). After manual curation of micrographs, reference-free particle picking was conducted using Laplacian-of-Gaussian filtering in RELION-3.0. Extracted particles were binned x4 (3.47 Å/pixel), and subjected to reference-free 2D classification with a 220 Å circular mask. Class averages that represented different views of the Fab bound Env-trimer and displayed secondary structural elements were selected (~641,000 particles) and an ab initio model was generated using cryoSPARC v2.12 (*Punjani et al., 2017*).

The generated volume was used as an initial model for iterative rounds of 3D classification (C1 symmetry, k = 6) in RELION 3.0. Particles corresponding to 3D class averages that displayed the highest resolution features were re-extracted unbinned (0.869 Å/pixel) and homogenously 3D-refined with a soft mask in which Fab constant domains were masked out, resulting in an estimated resolution of 4.1 Å according to gold-standard FSC (*Bell et al., 2016*). To further improve the resolution, particles were further 3D classified (k = 6, tau_fudge = 10), polished, and CTF refined. The final particle stack of ~404,000 particles refined to a final estimated resolution of 3.54 Å (C1 symmetry) according to gold-standard FSC.

To generate initial coordinates, reference models (gp120-gp41, PDB: 6UDJ; 8ANC195 Fab, 4PNM) were docked into the final reconstructed density using UCSF Chimera v1.13 (*Goddard et al., 2007*). For the Ab1485, initial coordinates were generated by docking a 10–1074 Fab model (PDB 4FQQ), which had been altered by removing Fab CDR loops, into the cryo-EM density at the V3/N332-glycan epitope. Prior to initial refinement, 10–1074 $V_H$-$V_L$ sequences were mutated to match Ab1485 and manually refined into density in Coot (*Emsley et al., 2010*). The full model was then refined into the cryo-EM maps using one round of rigid body refinement, morphing, and simulated annealing followed by several rounds of B-factor refinement in Phenix (*Adams et al., 2010*). Models were manually built following iterative rounds of real-space and B-factor refinement in Coot and Phenix with secondary structure restraints. Modeling of glycans was achieved by interpreting cryo-EM density at possible N-linked glycosylation sites in Coot. Validation of model coordinates was performed using MolProbity (*Chen et al., 2010*) and figures were rendered using UCSF Chimera or PyMOL (Version 1.5.0.4 Schrodinger, LLC). Buried surface areas and potential hydrogen bonds were determined as previously described (*Schoofs et al., 2019*).

## Surface Plasmon Resonance

SPR experiments were performed using a Biacore T200 instrument (GE Healthcare). RC1 SOSIP (*Escolano et al., 2019*), RC1 glycan-KO ³²⁴GAIA³²⁷SOSIP (*Escolano et al., 2019*) and BG505 SOSIP (*Escolano et al., 2019*) (were immobilized on a CM5 chip by primary amine chemistry (Biacore manual)). Flow cell one was kept empty and reserved as a negative control. A concentration series of 1485 Fab (3-fold dilutions from a top concentration of 100 nM) was injected at 30 μl/min for 60 s followed by a dissociation phase of 300 s. Binding reactions were allowed to reach equilibrium and $K_D$ values were calculated from the ratio of association and dissociation constants ($K_D = k_d/k_a$), which were derived using a 1:1 binding model that was globally fit to all curves in a data set. Flow cells were regenerated with 10 mM glycine pH 3.0 at a flow rate of 90 μl/min for 30 s.

## Virus challenge

A single dose (10 mg/kg body weight) of Ab1485 was infused intravenously into four Indian rhesus macaques (DH18, DH27, DH29, and DHAP). 24 hr following Ab infusion, these animals were

inoculated intrarectally with a high dose (1000 $TCID_{50}$) challenge of $SHIV_{AD8}$. Two control monkeys (FZH and JG7), receiving no Ab, were reported in a previous study (*Yamamoto, 2015*). A pediatric speculum was used to gently open the rectum and a 1 mL suspension of virus in a tuberculin syringe was slowly infused into rectal cavity. Blood was drawn regularly to monitor viral infection and serum neutralizing activity. Rhesus macaques were housed and cared for in accordance with Guide for Care and Use of Laboratory Animals Report number NIH 82–53 (Department of Health and Human Services, Bethesda, Maryland, 1985) in a biosafety level 2 National Institute of Allergy and Infectious Diseases (NIAID) facility. All animal procedures and experiments were performed according to LMM32E protocol approved by the Institutional Animal Care and Use Committee of NIAID, NIH.

### Authentication of cell lines

Cell lines were obtained from and authenticated by vendors or scientific collaborators. The cell lines were not contaminated by mycoplasma as determined by using the Lonza Mycoplasma Detection Kit.

### Analysis

MacVector v.17.0.2 was used for sequence analysis. Flow cytometry data were processed using FlowJo v.10.6.1and FCS EXPRESS. GraphPad Prism seven was used for data analysis. Immunoglobulin gene sequence AB1 files were converted to FASTQ format using SeqIO from Biopython (*Cock et al., 2009*). In the quality control step, non-determined and low-quality nucleotides were trimmed from both 5' and 3' ends of the sequence present in the FASTQ files using cutadapt v.2.3 software. IgBlast was used to identify immunoglobulin V(D)J genes and consequently the junction region, which was further used to define the Ig clones by Change-O toolkit v.0.4.5 (*Gupta et al., 2015*). Clones were defined by calculating and normalizing the hamming distance of the junction region and comparing it to a pre-defined threshold of 0.15.

## Acknowledgements

We thank members of the Bjorkman, Martin and Nussenzweig laboratories for discussions. Cryo-EM was performed in the Beckman Institute Resource Center for Transmission Electron Microscopy at Caltech with assistance from directors A Malyutin and S Chen. We thank J Vielmetter and the Beckman Institute Protein Expression Center at Caltech for protein production, John Moore (Weill Cornell Medical College) for the BG505 stable cell line and Rogier W Sanders (Amsterdam UMC) and Marit J van Gils (Amsterdam UMC) for providing BG505 SOSIP trimers. This work was supported by NIH Center for HIV/AIDS Vaccine Immunology and Immunogen Discovery (CHAVI-ID) 1UM1 AI100663-01 (to MCN), the National Institute of Allergy and Infectious Diseases (NIAID) HIVRAD P01 AI100148 (to MCN and PJB), the Bill and Melinda Gates Foundation Collaboration for AIDS Vaccine Discovery (CAVD) grant INV-002143 (to MAM, MCN, PJB), the Intramural Research Program of the NIAID, NIH (to MAM) and the Bill and Melinda Gates Foundation (CAVD) grant #OPP1146996 (to MSS). Additional support included the NIH K99/R00 grant (9694871) (to AE), the HHMI Hanna Gray Fellowship and the Postdoctoral Enrichment Program from the Burroughs Welcome Fund (to COB).

## Additional information

### Competing interests

Pamela J Bjorkman: Reviewing editor, *eLife*. The other authors declare that no competing interests exist.

### Funding

| Funder | Grant reference number | Author |
| --- | --- | --- |
| NIH Center for HIV/AIDS Vaccine Immunology and Immunogen Discovery | 1UM1 AI100663-01 | Michel C Nussenzweig |
| National Institute of Allergy | HIVRAD P01AI100148 | Pamela J Bjorkman |

| | | |
|---|---|---|
| and Infectious Diseases | | Michel C Nussenzweig |
| Bill and Melinda Gates Foundation | Collaboration for AIDS Vaccine Discovery INV-002143 | Malcolm Martin Pamela J Bjorkman Michel C Nussenzweig |
| National Institute of Allergy and Infectious Diseases | Intramural Research Program of the NIAID | Malcolm Martin |
| Bill and Melinda Gates Foundation | OPP1146996 | Michael S Seaman |
| National Institutes of Health | K99/R00 grant 9694871 | Amelia Escolano |
| Howard Hughes Medical Institute | Hanna Gray Fellowship | Christopher O Barnes |
| Burroughs Wellcome Fund | Postdoctoral Enrichment Program | Christopher O Barnes |

The funders had no role in study design, data collection and interpretation, or the decision to submit the work for publication.

### Author contributions

Zijun Wang, Data curation, Formal analysis, Investigation, Methodology; Christopher O Barnes, Conceptualization, Data curation, Formal analysis, Funding acquisition, Investigation, Methodology, Writing - original draft; Rajeev Gautam, Conceptualization, Data curation, Formal analysis, Investigation, Methodology, Writing - original draft, Writing - review and editing; Julio C Cetrulo Lorenzi, Christian T Mayer, Melissa Cipolla, Kristie M Gordon, Harry B Gristick, Yoshiaki Nishimura, Henna Raina, Methodology; Thiago Y Oliveira, Victor Ramos, Software; Anthony P West, Formal analysis; Michael S Seaman, Supervision, Funding acquisition, Methodology; Anna Gazumyan, Supervision, Methodology; Malcolm Martin, Michel C Nussenzweig, Conceptualization, Supervision, Funding acquisition, Writing - original draft, Project administration, Writing - review and editing; Pamela J Bjorkman, Supervision, Funding acquisition, Writing - original draft, Project administration, Writing - review and editing; Amelia Escolano, Conceptualization, Data curation, Formal analysis, Supervision, Funding acquisition, Investigation, Methodology, Writing - original draft, Project administration, Writing - review and editing

### Author ORCIDs

Victor Ramos http://orcid.org/0000-0001-7353-3420
Pamela J Bjorkman http://orcid.org/0000-0002-2277-3990
Amelia Escolano https://orcid.org/0000-0002-1945-2440

### Ethics

Animal experimentation: Rhesus macaques were housed and cared for in accordance with Guide for Care and Use of Laboratory Animals Report number NIH 82-53 (Department of Health and Human Services, Bethesda, Maryland, 1985) in a biosafety level 2 National Institute of Allergy and Infectious Diseases (NIAID) facility. All animal procedures and experiments were performed according to LMM32E protocol approved by the Institutional Animal Care and Use Committee of NIAID, NIH.

### Decision letter and Author response

Decision letter https://doi.org/10.7554/eLife.61991.sa1
Author response https://doi.org/10.7554/eLife.61991.sa2

# Additional files

### Supplementary files

• Transparent reporting form

## Data availability

Sequencing data has been provided for all the antibodies reported on the manuscript. Coordinates and corresponding 3D EM reconstructions for the Ab1485-BG505-8ANC195 trimer complex have been deposited in the PDB and EMDB, under accession numbers PDB: 7KDE and EMD: 22820.

The following datasets were generated:

| Author(s) | Year | Dataset title | Dataset URL | Database and Identifier |
|---|---|---|---|---|
| Barnes CO, Bjork-man PJ | 2020 | BG505 SOSIP.664 in complex with the V3-targeting rhesus macaque antibody 1485 and human gp120-gp41 interface antibody 8ANC195 | https://www.rcsb.org/structure/7KDE | RCSB Protein Data Bank, 7KDE |
| Barnes CO, Bjork-man PJ | 2020 | BG505 SOSIP.664 in complex with the V3-targeting rhesus macaque antibody 1485 and human gp120-gp41 interface antibody 8ANC195 | https://www.ebi.ac.uk/pdbe/entry/emdb/EMD-22820 | Electron Microscopy Data Bank, EMD-22820 |

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
