## [Decision Letter]

**Acceptance summary:**

This study describes a monoclonal antibody isolated from a SHIV-infected macaque that targets a common epitope of human HIV-specific antibodies. This is important because macaques are the benchmark model system for HIV vaccine testing, yet remarkably there have been few studies on what type of antibodies they make and whether they in fact are a relevant immunological preclinical model for vaccines design to elicit neutralizing antibodies. This paper begins to fill that gap and further shows that the antibody can block infection in macaques.

**Decision letter after peer review:**

Thank you for submitting your article "A Broadly Neutralizing Macaque Monoclonal Antibody Against the HIV-1 V3-Glycan Patch" for consideration by *eLife*. Your article has been reviewed by three peer reviewers, one of whom is a member of our Board of Reviewing Editors, and the evaluation has been overseen by Miles Davenport as the Senior Editor. The following individual involved in review of your submission has agreed to reveal their identity: Adam Dingens (Reviewer #2).

The reviewers have discussed the reviews with one another and the Reviewing Editor has drafted this decision to help you prepare a revised submission.

This is a pretty straightforward study of a monoclonal antibody isolated from a SHIV-infected macaque that shows reasonable breadth against HIV strains. The approach follows a very well-worn script for these studies in humans, but the novelty lies in the fact that the study was done in macaques. This is important because macaques are THE benchmark model system for HIV vaccine testing, yet remarkably there have been few studies on what type of antibodies they make and whether they in fact are a relevant immunological preclinical model for vaccines design to elicit neutralizing antibodies. This paper begins to fill that gap.

Summary:

Wang, Barnes, Gautam et al. describe the isolation and characterization of a monoclonal antibody (Ab1485) targeting the N332-supersite epitope from a macaque infected with SHIVAD8. Detailed structural work shows similarities and differences in the molecular interactions of Ab1485 with this epitope as compared to human bNAbs. They also show that Ab1485 protects against matched SHIVAD8 challenge in macaques. The main implication of this study is that macaques can develop N332-supersite antibodies that are structurally similar to some human infection-derived N332 supersite bNAbs. This suggests rhesus macaques could be a reasonable model for evaluating vaccination regimens that target this site, as the authors are doing in other work. Overall, the work is well done and relevant to the field but could be significantly improved without additional experiments.

Essential revisions:

There are a few limitations to the study that should be noted in the Discussion and the comparison with human V3- glycan Nab structures seems to be limited and could be expanded. For example, while the authors do a thorough job comparing the structure of Ab1485 to human N332 bNAbs, they do not adequately compare the functional activity and sequence similarity of Ab1485 to human N332 bNAbs. Authors should address these concerns and/or rephrase all statements such as "Thus, rhesus macaques develop anti-HIV-1 V3-glycan patch bNAbs that are related to human bNAbs" to include the limitation that these are structurally similar to/target similar epitopes to human bNAbs. They pick some select examples of N332 bNAb structures (Kong et al., 2013; Pancera et al., 2014; Freund et al., 2017). Why is this group important and does this nhp nab more resemble any of the other V3 bNAbs. This is also relevant to the Discussion (second paragraph). Do all other human V3 glycan bNAbs fit within these three examples or are there other ways these bNAbs can target antigen?

The authors used binding to BG505 to select 3 mabs to study. This is a rather important step in the process they used to decide which antibodies to study. What is the evidence that binding to BG505 trimer predicts breadth?

The Introduction is quite short and could benefit with some more detailed explanation of bNAb target epitopes and their development, in particular the ones targeting the V3-glycan site. Additionally, the authors could improve the introduction and contextualization of these results with prior literature. This would help readers who are not experts in macaque immune responses to SHIV infection (such as myself). What is known about the macaque antibody response to other SHIV strains beyond AD8 (and the single 14 animal study focused on in the Introduction)? From my limited knowledge, both Shingai et al., 2012, and the more recent 10.1101/2020.08.05.237693 warrant discussion. These are more relevant than the examples of low SHM that targets a glycan epitope, which does not really fit well anyway for this study.

The authors rightly point out that HIV vaccine concepts are tested in macaques without any knowledge of whether they make relevant responses to humans. Indeed, this is rather amazing and the relevance of the model seems quite important. In that regard, how relevant is the SHIV AD8 strain to vaccine efforts? Is the antigenicity of SHIVAD8 representative of HIV Env?. This adapted virus is relatively "open", is not neutralized by apex bNAbs, and may not be extremely relevant to protection against circulating strains. While this may not impact the N332 epitope extensively, the general antigenicity of this virus compared to circulating strains should be discussed when discussing the implications of using this macaque system for vaccine design. This should be discussed, potentially as a limitation.

It would be useful to see another V3 bNAb from human like 10-1074, tested in parallel, as comparison in Figure 1C. Data comparing the antibody neutralization to plasma neutralization should also be included.

---

## [Author Response]

Essential revisions:There are a few limitations to the study that should be noted in the Discussion and the comparison with human V3- glycan Nab structures seems to be limited and could be expanded. For example, while the authors do a thorough job comparing the structure of Ab1485 to human N332 bNAbs, they do not adequately compare the functional activity and sequence similarity of Ab1485 to human N332 bNAbs. Authors should address these concerns and/or rephrase all statements such as "Thus, rhesus macaques develop anti-HIV-1 V3-glycan patch bNAbs that are related to human bNAbs" to include the limitation that these are structurally similar to/target similar epitopes to human bNAbs.

Thanks to the reviewers for these suggestions. We have added neutralization data for a series of human V3-glycan bNAbs to Figure 1C to compare their potency and breadth with those of Ab1485. Also, we have discussed in the text the similarities/differences between Ab1485 and these human V3-glycan bNAbs (Results).

They pick some select examples of N332 bNAb structures (Kong et al., 2013; Pancera et al., 2014; Freund et al., 2017). Why is this group important and does this nhp nab more resemble any of the other V3 bNAbs. This is also relevant to the Discussion (second paragraph). Do all other human V3 glycan bNAbs fit within these three examples or are there other ways these bNAbs can target antigen?

We chose three potent human bNAbs (PGT128, BG18, and 10-1074) for structural comparisons with Ab1485 (Figure 3D-F) because available structures of Env trimers with these bNAbs showed that they target the V3-glycan patch using different angles of approach. We did not include a figure comparing Ab1485 with PGT135, which adopts yet another binding pose, because this antibody is less broad and potent than the others, and because we had previously compared its binding to the binding of BG18 (Figure 2E-F) in Freund et al., 2017.

The authors used binding to BG505 to select 3 mabs to study. This is a rather important step in the process they used to decide which antibodies to study. What is the evidence that binding to BG505 trimer predicts breadth?

This is a good point. We used BG505 binding in ELISA to select antibodies for further analysis because it was one of the baits used to isolate Env-binding B cells by FACS. Binding to BG505 in ELISA was used to confirm that the isolated antibodies were Env-specific. Moreover, BG505.664 has previously been shown to bind to many human broadly neutralizing antibodies and to show reduced binding to non-neutralizing antibodies (Sanders et al., 2013). We have clarified this point in the revised manuscript (subsection “Ab1485 isolated from macaque CE8J is potently neutralizing”).

The introduction is quite short and could benefit with some more detailed explanation of bNAb target epitopes and their development, in particular the ones targeting the V3-glycan site. Additionally, the authors could improve the introduction and contextualization of these results with prior literature. This would help readers who are not experts in macaque immune responses to SHIV infection (such as myself). What is known about the macaque antibody response to other SHIV strains beyond AD8 (and the single 14 animal study focused on in the Introduction)? From my limited knowledge, both Shingai et al., 2012 and the more recent 10.1101/2020.08.05.237693 warrant discussion. These are more relevant than the examples of low SHM that targets a glycan epitope, which does not really fit well anyway for this study

Thanks to the reviewers for their suggestions to improve the manuscript. We have extended the Introduction and Discussion sections including the relevant literature.

The authors rightly point out that HIV vaccine concepts are tested in macaques without any knowledge of whether they make relevant responses to humans. Indeed, this is rather amazing and the relevance of the model seems quite important. In that regard, how relevant is the SHIV AD8 strain to vaccine efforts? Is the antigenicity of SHIVAD8 representative of HIV Env?. This adapted virus is relatively "open", is not neutralized by apex bNAbs, and may not be extremely relevant to protection against circulating strains. While this may not impact the N332 epitope extensively, the general antigenicity of this virus compared to circulating strains should be discussed when discussing the implications of using this macaque system for vaccine design. This should be discussed, potentially as a limitation.

Thank you to the reviewers for raising these questions. SHIV_AD8-EO_ is not neutralized by the PG9 and PG16 apex bNAbs but is neutralized by the PGT145 apex bNAb (Immunity 50:567; mBio 10:e01255-19). Andres Finzi has unpublished data showing that unlike SHIV162P3, SHIV_AD8-EO_ has a closed Trimer conformation. One can question whether the envelopes of “circulating” or “founder” strains of HIV-1 for human infection are relevant for SHIV studies of macaques – the key limiting factor for the latter appears to be usage of monkey CD4 for virus entry. We have indicated that SHIVAD8 exhibits a Tier 2 neutralization sensitivity phenotype in the Introduction.

It would be useful to see another V3 bNAb from human like 10-1074, tested in parallel, as comparison in Figure 1C. Data comparing the antibody neutralization to plasma neutralization should also be included.

We have updated Figure 1C to include neutralization data for several human V3-glycan bNAbs. Information about the neutralization activity detected in the plasma from this macaque can be found in Walker et al., 2011. We have included this information in the Results.